# Effects of Gestational Diabetes Mellitus on Fetal Cardiac Morphology

**DOI:** 10.3390/medsci12040073

**Published:** 2024-12-14

**Authors:** Esra Söylemez, Sermet Sağol

**Affiliations:** 1Department of Obstetrics and Gynecology, Mardin Education and Research Hospital, Mardin 47100, Turkey; 2Department of Perinatology, Ege University, İzmir 35000, Turkey; sermet.sagol@ege.edu.tr

**Keywords:** BMI (body mass index), OGTT (oral glucose tolerance test), GDM (gestational diabetes mellitus), echocardyography, fetal cardiac morphology

## Abstract

Objective: This study aims to investigate the possible effects of gestational diabetes mellitus (GDM) on fetal heart structure and the relationship of this effect with maternal blood sugar control. Materials and Methods: In this cross-sectional study, 19 women with GDM at 24–36 weeks of gestation (case group) and 21 healthy pregnant women at the same weeks of gestation (control group) were examined. Fetal heart structure was evaluated by ultrasonography; interventricular septum (IVS) thickness, right and left ventricular sphericity indices, global sphericity index (GSI) and cardio-thoracic ratio were also measured. In addition, mothers’ HbA1c values (an indicator of blood sugar control) were recorded. Result: An increase in IVS thickness was observed in the fetuses of mothers with GDM. A more rounded trend was observed in the right ventricular structure, but this did not create a significant difference. No significant relationship was found between maternal blood sugar control and fetal heart structure. Conclusions: This study examined the effects of gestational diabetes on fetal cardiac morphology and the relationship of this effect with maternal glycemic control. Babies of mothers with GDM had a significantly thicker interventricular septum. A more rounded trend was detected in the right ventricular structure. However, this change was not found to be statistically significant. In addition, no significant correlation was found between maternal glycemic control and fetal cardiac morphology.

## 1. Introduction

Gestational diabetes mellitus (GDM) is a common metabolic disorder characterized by high blood sugar during pregnancy and significantly increases perinatal morbidity and mortality [1]. The pathogenesis of diabetes mellitus is quite complex and is associated with insulin resistance and various hormonal changes [2]. Pregnancy is a condition that causes insulin resistance and high insulin due to its nature, and contributes to the development of diabetes as a primary factor.

It is well known that maternal hyperglycemia adversely affects fetal development, particularly with notable effects on the fetal heart [3,4]. Animal studies have shown that maternal hyperglycemia can lead to remodeling in fetal myocardial fibers, which may increase the risk of cardiovascular diseases later in life [3]. It has been revealed that the occurrence of these complications related to gestational diabetes mellitus and the severity of their clinical reflection are directly related to the mother’s blood glucose control level [5].

There are several studies in the literature investigating the effects of GDM on the fetal cardiovascular system; however, these studies generally focus on pre-gestational diabetes and cardiac functions [6]. In this study, the effects of GDM on fetal heart structure and the relationship between these effects and maternal blood sugar control were examined. Specifically, changes in the heart structure of fetuses of mothers with GDM and the correlation of these changes with maternal blood sugar control were evaluated. This study may provide important insights into understanding the effects of GDM on fetal cardiac morphology and developing potential intervention strategies.

## 2. Materials and Methods

Our study was designed in accordance with the Declaration of Helsinki. Ethical approval of our study was obtained from the Ege University ethics committee (Study Ethics Committee no: 22-2T/28). The patients included in the study were informed, and written informed consent was obtained.

Pregnant women with GDM between 24 and 36 weeks of gestation were selected for this study. The research included patients who applied to the Perinatology Polyclinic of the Department of Obstetrics and Gynecology of Ege University between February 2022 and June 2022 and were diagnosed with GDM between 24 and 28 gestational weeks. A total of 40 patients, including 19 diabetic pregnant women and 21 control groups, were selected according to the single-step 75 g oral glucose tolerance test (OGTT). This test includes three-step glucose level measurements, and in this study, IADPSG’s cut-off values were taken as basis.

All examinations were performed by the same, single observer, with the patient in the supine position, using the GE Voluson E8 ultrasound device’s 2–9 MHz wide-band convex abdominal probe (C2-9-D) and obstetric transducer. Cardiac structure was evaluated using gray-scale and fetal M mode. Images were taken for two or three seconds while the mother was holding her breath and the fetus was not moving, in a 4-chamber view of the fetal heart, perpendicular to the long axis or as close to the perpendicular position as possible. Measurements were made end-diastolically, and the interventricular septum (IVS) thickness (with M-mode), left ventricular (LV) and right ventricular (RV) sphericity index (ratio of the apico-basal ventricular length to the transverse length at the level of the relevant atrio-ventricular valve) were measured; the global sphericity index (GSI) was calculated by dividing the basal–apical length of the fetal heart at the end of diastole by the transverse width at the end of diastole. In addition, the fetal heart rate was recorded, and the fetal cardiothoracic ratio (fetal heart circumference divided by the fetal thoracic circumference) was calculated. HbA1c(%) was recorded as biochemical parameter.

The inclusion criteria for the GDM group were singleton pregnancies of mothers aged 18 years and over, a positive diagnosis of GDM at Ege University Hospital, and a gestational age between 24 and 36 weeks. The exclusion criteria for the GDM and control groups were fetuses with structural or chromosomal anomalies; fetal tachycardia (FHR > 160 beats/minute); maternal smoking during pregnancy; twin or multiple pregnancies; inadequate ultrasonographic images; any maternal disease, including chronic hypertension, diabetes mellitus, human immunodeficiency virus or hepatitis infection before pregnancy, thyroid disease, or other uncategorized diseases that would impair fetal heart morphology and function; fetuses with congenital heart disease, arrhythmia or malformations involving other organs; fetuses with intrauterine growth retardation (EFW < 10 p); and mothers’ refusal to give written informed consent.

### Statistical Analysis

The statistical analysis of the data were conducted using the SPSS 22.0 software package. The Kolmogorov–Smirnov test was employed to assess whether the data followed a normal distribution. For data with a normal distribution, results are expressed as mean ± standard deviation, whereas for data not following a normal distribution, results are presented as the median (minimum–maximum). Comparisons between groups were made using the independent samples *t*-test for normally distributed data, and the Mann–Whitney U test for data not normally distributed. Pearson correlation analysis was used for correlation assessment. A significance level of *p* < 0.05 was considered statistically significant.

## 3. Results

A total of 40 pregnant women were included in the study, consisting of 19 GDM patients as the case group and 21 healthy pregnant women as the control group. These participants were selected from pregnant women who presented at the Perinatology Clinic of Ege University’s Department of Obstetrics and Gynecology between February and June 2022, following the application of various exclusion criteria.

The age range of participants in the study was 19–41 years, with a mean age of 29 years. Among all cases, the shorted gestational week was 25 and the longest gestational week was 35. The mean BMI of the participants was found to be 26.6. The mean HbA1c value was 5.02 across the entire study population. In total, 40% of all participants have a family history of diabetes. In the GDM group, 63.2% of the pregnant women managed their blood sugar levels through diet, while 36.8% required insulin therapy. No patients were using oral antidiabetic medications. Blood glucose regulation thresholds were determined according to ACOG (American College of Obstetricians and Gynecologists) recommendations (Table 1).

There was no significant difference between the GDM and control groups in terms of age and gestational week. However, BMI and HbA1c values were significantly higher in the GDM group (*p* = 0.033 and *p* = 0.049, respectively). It was observed that 42.1% of pregnant women with GDM had a family history of diabetes in first-degree relatives (mother, father, and siblings). The fetal heart rate was similar in both groups. Ultrasound measurements revealed that IVS thickness was significantly increased in the GDM group (*p* = 0.048). While there was no significant difference in the global sphericity index (GSI) and LV sphericity indices, the RV sphericity index was lower in the GDM group, but this difference was not statistically significant. The cardiothoracic ratio was compared between the case and control groups by matching gestational weeks and it was significantly lower in the GDM group (*p* < 0.001) (Table 2).

According to the data in the study, there was no significant correlation between the HbA1c value of fetal cardiac morphological ultrasonographic measurements in the GDM group (Table 3).

## 4. Discussion

Gestational diabetes mellitus (GDM) is a common metabolic disorder encountered during pregnancy that negatively affects maternal and fetal health [7]. It is well established that elevated blood sugar levels (hyperglycemia) in the mother can adversely affect fetal development, particularly the fetal heart [8,9]. In this study, the effects of GDM on fetal cardiac morphology and the relationship of these effects with maternal glycemic control were examined.

Our most important finding is that there is a significant increase in interventricular septum (IVS) thickness in the fetuses of mothers with GDM, even when blood sugar levels are well controlled. This finding is consistent with previous studies, such as those of Garg et al. [10] and Balli et al. [11], which also showed an increase in heart wall thickness, particularly in the IVS, in the fetuses of mothers with GDM despite glycemic control. Our study supports these findings, reinforcing the notion that GDM may have adverse effects on fetal cardiac development. This increase in IVS thickness suggests that fetal cardiac health should be closely monitored in GDM pregnancies and that IVS thickness may be an important marker for the early detection of potential cardiac complications.

Another important contribution of our study to the literature is that it shows a significantly lower cardiothoracic ratio (CTR) in fetuses of mothers with GDM. This finding contradicts some studies in the literature. For example, Lehtoranta et al. [12] and Gandhi et al. [13] reported that hyperglycemia (high blood sugar) leads to fetal heart enlargement and an increase in CTR. However, in our study, CTR was found to be lower in the GDM group compared to the control group. We do not yet know the exact reason for this difference. However, the fact that only one observer was involved in our study and the possibility of errors in the imaging/measurement process may have contributed to this difference. This finding suggests that there is still much to learn about how GDM affects fetal cardiac development and that further research is needed in this area.

In a study conducted by Wang et al. [14], it was reported that the fetal heart in the GDM group had a rounder shape compared to the control group. However, in this study, the global sphericity index (GSI) was found to be similar in both groups, and no significant difference was detected between the groups. This discrepancy could be due to the fact that most of the women with GDM in this study had good glycemic control achieved through diet, without the need for medication. Nevertheless, consistent with Wang et al.’s [14] study, this study also found that the right ventricle (RV) sphericity index was significantly lower than the left ventricle (LV) sphericity index, indicating that the RV had a more spherical shape. This finding is also in line with the results of studies by Vore et al. [15] and Patey et al. [16].

In our study, we also investigated the relationship between fetal echocardiographic parameters and maternal glycemic control. Our findings showed no significant relationship between fetal myocardial thickness and maternal HbA1c levels (an indicator of long-term blood sugar control). This finding is partially inconsistent with some studies in the literature. For example, Tejaswi et al. [17] reported a significant relationship between maternal glycemic control and fetal myocardial thickness in their study. This discrepancy may be due to factors such as the smaller sample size in our study, the use of HbA1c as a single biochemical marker, and the fact that most of the cases included in the study were controlled by diet alone. This finding suggests that the effect of glycemic control on fetal cardiac development in GDM pregnancies is complex and requires further investigation.

### Limitations

The limitations of this study include the small sample size, the use of a single glycemic control parameter, and the lack of subgroup analysis. Future research should be supported by larger, multicenter, prospective randomized controlled trials to confirm and expand upon these findings.

## 5. Conclusions

This study was conducted between February and June 2022 at Ege University, involving 19 pregnant women with GDM and 21 healthy pregnant women at 24–36 weeks of gestation. Fetal heart structure was evaluated using ultrasonography, and measurements of interventricular septum (IVS) thickness, ventricular sphericity indices, global sphericity index (GSI), cardiothoracic ratio, and HbA1c levels were recorded.

In the GDM group, BMI and HbA1c values were found to be significantly higher compared to the control group. If we look at the ultrasonographic measurements, IVS thickness was significantly increased in the GDM group compared to the control group. While the RV sphericity index was lower in the GDM group, this difference was not statistically significant. The cardiothoracic ratio was significantly lower in the GDM group. No significant correlation was found between fetal cardiac measurements and HbA1c levels in the GDM group.

## Figures and Tables

**Table 1 medsci-12-00073-t001:** Distribution characteristics of demographic data and patient measurements of all participants.

			All participants
			Mean ± SD (Min–Max)
Age Mean ± SD (Min–Max)			29.1 ± 4.6 (19–41)
BMI Mean ± SD (Min–Max)			26.6 ± 4.3 (18.8–37.8)
Family history of diabetesn (%)			16 (40.0)
Gestational week Mean ± SD (Min–Max)			30.4 ± 2.97 (25–35)
OGTT n (%)		Normal	21 (52.5)
		GDM	19 (47.5)
HbA1c Mean ± SD (Min–Max)			5.02 ± 0.38 (4.4–6)
IVS thickness Mean ± SD (Min–Max)			5.07 ± 1.08 (2.9–7.5)
LV sphericity index Mean ± SD (Min–Max)			1.85 ± 0.48 (1–2.9)
RV sphericity index Mean ± SD (Min–Max)			1.71 ± 0.39 (0.7–3)
Global sphericity index Mean ± SD (Min–Max)			1.27 ± 0.15 (0.9–1.6)
CTR Mean ± SD (Min–Max)			0.43 ± 0.11 (0.2–0.6)
GDM therapyn (%)	Diet		12 (63.2)
	İnsulin		7 (36.8)
FHR Mean ± SD (Min–Max)			142.8 ± 7.9 (124–160)

BMI: body mass index; IVS: interventricular septum; LV: left ventricle; RV: right ventricle; CTR: cardiothoracic ratio; FHR: fetal heart rate; GDM: gestational diabetes mellitus; OGTT: oral glucose tolerance test.

**Table 2 medsci-12-00073-t002:** Comparisons of demographic data and patient measurements by case and control groups.

	Normal	GDM	
	Mean ± SD Min–Max (Median)	Mean ± SD Min–Max (Median)	*p* *
Age	Mean ± SD 28.4 ± 4.5 Min–Max (Median) 19–37 (29)	29.9 ± 4.7 23–41 (29)	0.320
BMI	Mean ± SD 25.2 ± 3.3 Min–Max (Median) 18.8–30.1 (25)	28.1 ± 4.83 22–37.8 (26.9)	0.033
Family history of diabetes n (%)	8 (38.1)	8 (42.1)	0.796
Gestational week	Mean ± SD 30.81 ± 3.04 Min–Max (Median) 25–35 (31)	30.0 ± 2.91 26–35 (30)	0.396
HbA1c	Mean ± SD 4.9 ± 0.26 Min–Max (Median) 4.6–5.5 (4.9)	5.15 ± 0.45 4.4–6 (5)	0.049
IVS thickness	Mean ± SD 4.75 ± 1.01 Min–Max (Median) 2.9–7.5 (4.54)	5.42 ± 1.07 3.1–7 (5.35)	0.048
LV sphericity index	Mean ± SD 1.81 ± 0.52 Min–Max (Median) 1–2.9 (1.75)	1.89 ± 0.45 1.3–2.8 (1.74)	0.587
RV sphericity index	Mean ± SD 1.73 ± 0.42 Min–Max (Median) 1.2–3 (1.7)	1.68 ± 0.37 0.7–2.3 (1.75)	0.742
Global sphericity index	Mean ± SD 1.27 ± 0.13 Min–Max (Median) 1–1.5 (1.29)	1.27 ± 0.17 0.9–1.6 (1.27)	0.998
CTR	Mean ± SD 0.50 ± 0.04 Min–Max (Median) 0.5–0.6 (0.49)	0.36 ± 0.13 0.2–0.5 (0.42)	<0.001
FHR	Mean ± SD 142.8 ± 9.1 Min–Max (Median) 124–160 (145)	142.8 ± 6.6 130–155 (143)	0.991

BMI: body mass index; IVS: interventricular septum; LV: left ventricle; RV: right ventricle; CTR: cardiothoracic ratio; FHR: fetal heart rate; GDM: gestational diabetes mellitus, *: Student *t* test.

**Table 3 medsci-12-00073-t003:** Investigation of the relationship between HbA1c and fetal cardiac indices.

N = 19		HbA1c	IVS Thickness	LV Sphericity Index	RV Sphericity Index	Global Sphericity Index	CTR
IVS thickness	r	0.149					
	*p*	0.542					
LV sphericity index	r	−0.419	0.329				
	*p*	0.074	0.169				
RV sphericity index	r	−0.321	0.071	0.29			
	*p*	0.180	0.772	0.228			
Global sphericity index	r	−0.051	−0.201	0.226	−0.021		
	*p*	0.836	0.408	0.353	0.933		
CTR	r	0.041	0.187	0.134	0.339	−0.35	
	*p*	0.867	0.443	0.585	0.156	0.142	
FHR	r	0.052	−0.166	−0.063	0.118	0.317	−0.335
	*p*	0.832	0.498	0.797	0.632	0.187	0.161

N: number; r: Pearson correlation coefficient; *p*: *p*-value; IVS: interventricular septum; LV: left ventricle; RV: right ventricle; CTR: cardiothoracic ratio; FHR: fetal heart rate.

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
