# Peer review of "Effects of Gestational Diabetes Mellitus on Fetal Cardiac Morphology"

_medsci, 2024, doi:10.3390/medsci12040073_

Round 1
Reviewer 1 Report
Comments and Suggestions for Authors
1. line 16 - double space (check whole manuscript)
2. line 25 - bady? please check the whole manuscript for grammar
3. line 39 - where is the reference for this animal study?
4. the introduction section needs to be improved, you only have 5 references. The introduction should explain to the reader why the research was conducted in the first place and ensure coherence from the first to the last paragraph.
5. line 43 and line 50 are almost the same, please just leave aim in the introduction
6. line 54 - add ethical approval
7. improve all tables (dont use coma for decimal place and explain in each line how data are presented (mean or median or proportion)
8. start discussion with your main findings and how is this relevant in the field or how it adds to the exsisting body of literature
9. line 171-208 - try to compare cited references with your results in two paragraphs without extraneous information
10. add limitation section at the end of the discussion
Author Response
Dear Reviewer
Thank you for your valuable comments.
- line 16 - double space (check whole manuscript)
Response to the comment 1: The whole manuscript has been checked and corrected
- line 25 - bady? please check the whole manuscript for grammar
Response to the comment 2: The whole manuscript has been checked and corrected
- line 39 - where is the reference for this animal study?
Response to the comment 3: Animal study reference added.
REFERANCE 3. Cohen K, Waldman M, Abraham NG, et al.: Caloric restriction ameliorates cardiomyopathy in animal model of diabetes. Exp Cell Res. 20171, 350:147-153. 10.1016/j.yexcr.2016.11.016
- the introduction section needs to be improved, you only have 5 references. The introduction should explain to the reader why the research was conducted in the first place and ensure coherence from the first to the last paragraph.
Response to the comment 4: The introduction section has been revised in line with your valuable suggestions, and the number of references has been increased.
REFERANCE 5. Crowther CA, Hiller JE, Moss JR, McPhee AJ, Jeffries WS, Robinson JS; Australian Carbohydrate Intolerance Study in Pregnant Women (ACHOIS) Trial Group. Effect of treatment of gestational diabetes mellitus on pregnancy outcomes. N Engl J Med. 2005 Jun 16;352(24):2477-86. doi: 10.1056/NEJMoa042973. Epub 2005 Jun 12. PMID: 15951574.
- line 43 and line 50 are almost the same, please just leave aim in the introduction
Response to the comment 5: Line 50 deleted
- line 54 - add ethical approval
Response to the comment 6: ethical approval added
- improve all tables (dont use coma for decimal place and explain in each line how data are presented (mean or median or proportion)
Response to the comment 7: Tables were arranged according to the suggestions
- start discussion with your main findings and how is this relevant in the field or how it adds to the exsisting body of literature
Response to the comment 8: The discussion section has been rearranged in accordance with the suggestions. References have been reordered.
- line 171-208 - try to compare cited references with your results in two paragraphs without extraneous information
Response to the comment 9: The discussion section has been shortened in accordance with the suggestions
- add limitation section at the end of the discussion
Response to the comment 10: The limitation section has been moved to the end of the discussion
Thank you very much for your precious time.
Best regards
Reviewer 2 Report
Comments and Suggestions for Authors
The authors studied the correlation between gestational diabetes mellitus and fetal cardiac morphology. The study validated one of the previous observations that fetuses have a significantly thicker interventricular septum when their mothers have gestational diabetes mellitus. The study design is appropriate. A cohort of 19 cases and 21 controls is relatively small. The result presentation is OK, even though Table 1 is less relevant. The amount of work is relatively small for an essay.
Author Response
Dear Reviewer
Thank you for your valuable comments.
The authors studied the correlation between gestational diabetes mellitus and fetal cardiac morphology. The study validated one of the previous observations that fetuses have a significantly thicker interventricular septum when their mothers have gestational diabetes mellitus. The study design is appropriate. A cohort of 19 cases and 21 controls is relatively small. The result presentation is OK, even though Table 1 is less relevant. The amount of work is relatively small for an essay.
Response to the comment: Due to the presence of many different hospitals in the province where we conducted our study and the fact that our clinic is a tertiary center, the number of patients admitted during the study period was limited. Among the patients who applied, there were those who did not give their consent or gave up participating in the study. Due to the design of the study, the selection of pregnant women between 24-36 weeks was another factor that reduced the number.
Thank you very much for your precious time.
Best regards
Reviewer 3 Report
Comments and Suggestions for Authors
This article describes a relation between gestational diabetes and cardiac morphology.
The numbers are not big, but the findings for instance the increase in IVS thickness in the fetuses confirms findings of others.
What I don’t understand is the significant smaller cardio thoracic ratio. Did you control the ratio for gestational age?? I couldn’t find a word in your abstract about that finding.
When you explain that finding or skip it , I am happy with the article.
In table 3, what means r and P : controls and patients?? That is not made clear in the paper.
There is a typo in the keywords: BMI (body not bady mass index.
Author Response
Dear Reviewer
Thank you for your valuable comments.
This article describes a relation between gestational diabetes and cardiac morphology.
The numbers are not big, but the findings for instance the increase in IVS thickness in the fetuses confirms findings of others.
What I don’t understand is the significant smaller cardio thoracic ratio. Did you control the ratio for gestational age?? I couldn’t find a word in your abstract about that finding.
When you explain that finding or skip it , I am happy with the article.
Response to the comment 1: According to the study design, the control and case groups consisted of women with similar gestational weeks. Cardiothoracic ratio was compared between the case and control groups by matching gestational weeks. An explanatory sentence has been added to the results section related to this.
In table 3, what means r and P : controls and patients?? That is not made clear in the paper.
Response to the comment 2: r: Pearson correlation coefficient, p: p-value. It has been added to the explanation section of Table 3.
There is a typo in the keywords: BMI (body not bady mass index.
Response to the comment: corrected
Thank you very much for your precious time.
Best regards
Round 2
Reviewer 1 Report
Comments and Suggestions for Authors
I believe it is now acceptable for publication
Author Response
I believe it is now acceptable for publication
Response to the comment:
Dear Reviewer,
Thank you for your valuable comments.
Best regards
Reviewer 2 Report
Comments and Suggestions for Authors
There is no significant change. I have no additional comments.
Author Response
Dear Reviewer,
Thank you for your valuable comments.
Comment: There is no significant change. I have no additional comments.
Response to the comment: First, we understand your concern that our working group is small. However, we would like to point out that this number is the highest that can be reached under current conditions, considering that there are many different hospitals in the province where our study was conducted, the study was carried out in a tertiary center and included patients within a certain range of gestational weeks. We would also like to emphasize that the results of our study are statistically significant and in line with previous studies.
Thank you very much for your precious time.
Best regards
Reviewer 3 Report
Comments and Suggestions for Authors
I am satisfied, the revised article can be accepted
Author Response
I am satisfied, the revised article can be accepted
Response to the comment:
Dear Reviewer,
Thank you for your valuable comments.
Best regards